# Quantitative CPP Evaluation from Risk Assessment Using Integrated Process Modeling

**DOI:** 10.3390/bioengineering6040114

**Published:** 2019-12-13

**Authors:** Daniel Borchert, Thomas Zahel, Yvonne E. Thomassen, Christoph Herwig, Diego A. Suarez-Zuluaga

**Affiliations:** 1Exputec GmbH, Mariahilferstraße 88a/1/9, 1070 Vienna, Austria; daniel.borchert@exputec.com (D.B.); thomas.zahel@exputec.com (T.Z.); 2Intravacc, Antonie van Leeuwenhoeklaan 9, 3721 MA Bilthoven, The Netherlands; yvonne.thomassen@intravacc.nl (Y.E.T.); diego.suarez@intravacc.nl (D.A.S.-Z.); 3Research Area Biochemical Engineering, Vienna University of Technology, Gumpendorferstraße 1a, 1060 Vienna, Austria

**Keywords:** integrated process modeling, OOS estimation, simulation based on risk assessment, process knowledge, potential critical process parameter (pCPP) assessment, severity contribution, occurrence contribution

## Abstract

Risk assessments (RAs) are frequently conducted to assess the potential effect of process parameters (PPs) on product quality attributes (e.g., a critical quality attribute (CQA)). To evaluate the PPs criticality the risk priority number (RPN) for each PP is often calculated. This number is generated by the multiplication of three factors: severity, occurrence, and detectability. This mathematical operation may result in some potential errors due to the multiplication of ordinal scaled values and the assumption that the factors contribute equally to the PPs criticality. To avoid these misinterpretations and to assess the out of specification (OOS) probability of the drug substance, we present a novel and straightforward mathematical algorithm. This algorithm quantitatively describes the PPs effect on each CQA assessed within the RA. The transcription of severity and occurrence to model effect sizes and parameters distribution are the key elements of the herein developed approach. This approach can be applied to any conventional RA within the biopharmaceutical industry. We demonstrate that severity and occurrence contribute differently to the PP criticality and compare these results with the RPN number. Detectability is used in a final step to precisely sort the contribution of each factor. To illustrate, we show the misinterpretation risk of the PP critically by using the conventional RPN approach.

## 1. Introduction

Risk assessment (RA) is a famous tool for risk evaluation, within a biopharmaceutical process, to evaluate the likely reason for not meeting drug substance (DS) specification. The main focus of the RA is to identify potential critical process parameters (pCPPs) affecting a particular critical quality attribute (CQA). As a result, the identified CPPs can be further assessed using a design of experiment (DoE) approach. Considering the International Conference of Harmonization (ICH) Q11 guideline, a CQA is defined as “A physical, chemical, biological or microbial property or characteristic that should be within an appropriate limit range or distribution to ensure the desired product quality” [1]. Hence, it is crucial to investigate their influencing critical process parameters (CPPs) with the utmost precision and set an appropriate control strategy [2,3]. 

For biopharmaceutical process development, the ICH Q9 Guideline addresses risk management, and suggests conducting a RA for each CQA individually. A risk is defined as a function of the severity of consequence and probability of occurrence [3]. Due to its systematic approach for risk management, this guideline additionally lays the basis for the quality by design approach [4,5,6], which has been implemented in the biopharmaceutical industry for years. 

As a RA method, a failure mode and effects analysis (FMEA) tool aims to determine how a process can fail and evaluate the effect of this failure on the product [7,8]. This is determined via the risk priority number (RPN) approach which ranks the assessed risk of potential failure mode. The RPN is calculated by multiplying three factors: severity (S), occurrence (O), and detectability (D). Where S represents the severity of failure mode, O the probability of the failure, and D the likelihood of detecting the failure [6]. Resulting from its extensive use for many different applications, the RPN approach has been the most used assessment approach for years [9,10]. 

Although widespread in the industry, it is often reported that the RPN approach is not the best assessment tool for such an evaluation [10,11]. This statement is supported by the following reasons:Multiplication of ordinal scaled valuesOrdinal data tells us the ranking of a property, but only relative to each other. Therefore, from a mathematical point of view, a multiplication of different ordinal scales values is not correct [7].Gaps within the scaleDepending on the number of levels within S, O, and D that are used to obtain the RPN, many holes within the resulting RPN scale occur. The numbers generally cover a range from 1 to 1000 or from 1 to 125. However, 88% of the RPN scale is empty [7,8].RPNs with the same scores but derived differentlyThe S, O, and D factors always have different weights to their purpose and should not be considered as equally weighted. An assessment where S = 2, O = 7, and D = 4 results in an RPN value of 56. An equal RPN number can be generated by an assessment of S = 7, O = 4, and D = 2. Even though the RPN number is identical in both cases, the impact onto the CQA might be different due to its different factor estimation [11,12].Big influence by small changesSmall changes in one of the factors result in a significant shift in the final RPN. This shift also depends on the multiplication nature of the RPN calculation, which may be avoided for risk evaluation [13].

Several studies have discussed improvement strategies of the commonly used RPN approach [14]. To evaluate the weights of certain factors assessed in the FMEA, new methods like the hierarchical analytical process [15,16] or the Kano model [17] have been introduced. Furthermore, grey relation analysis [11,18,19] or the technique for order of preference by the similarity of ideal solution (TOPSIS) [20,21] are often combined into the FMEA to tune the RPN assessment. To decrease the term of uncertainty in many RA approaches, mitigate the ambiguity, and improve reliability, fuzzy logic algorithms have also been applied [18,19,20]. Since these tools tune the RPN evaluation, we present an approach which mathematically describes the impact on the out of specification (OOS) events without using the RPN.

All multi attribute decision making approaches mentioned above are based on sophisticated mathematical algorithms, which are hard to interpret for many process experts within the biopharmaceutical industry. Therefore, and for continuous RA improvement, we present a simple linearization procedure of the RA based just on expert assessment. For this approach, an FMEA is conducted as usual using S, O, and D for each PP. 

The herein developed tool uses a linearization approach to evaluate the PP criticality. For this approach, the S estimates are changed to model effect size and the O to model process parameter probability distribution. These parameters are further combined with start CQA values to get the linearization equation (Figure 1b2). We present an assessment approach which simulates the expected increase of OOS events associated with each PP evaluated within the RA. The holistic process evaluation is conducted by applying a Monte Carlo simulation [22], connecting each individual unit operation of the biopharmaceutical process. Such a simulation strategy is called an integrated process model (IPM) and enables the simulation of the process outputs from a holistic point of view across the unit operation. 

Zahel et al. [23] recently introduced the methodology of IPM into the biopharmaceutical industry and used manufacturing data to simulate the OOS by using the IPM approach. For our simulation, no experimental data are needed, and the herein developed IPM simulation uses all the information available within the RA. This simulation can be performed for any process once an RA is conducted and can be used either at early process development or for late process assessment.

For algorithm evaluation, in-silico data was generated and used. Furthermore, results based on a real RA were used to demonstrate the robustness of the tool and to verify the applicability on different types of CQAs. 

## 2. Materials and Methods 

Here, we summarize the format of the RA as well as the assumptions that must be met to conduct the simulation:Fulfill an FMEA (Section 2.1). As a simulation basis, an RA should be performed for each unit operation with respect to each CQA within the process.Estimate critical loss of CQA (or critical delta of CQA) per unit operation (Section 2.2). The critical loss of CQA resulting in OOS for each unit operation needs to be defined.Interpretation of S assessment (Section 2.3). We assume that the setpoint of the PP is chosen as the best operational condition. It is also assumed that deviating from the operation setpoint correlates with impact on the CQA. The S value describes the size of the effect affecting the OOS, and high S values indicate a higher contribution to a loss in OOS when deviating from the process setpoint. This assumption mainly considers the main effects of the PP.Interpretation of O assessment (Section 2.4). We assume the process expert evaluates the change of each PP within the judged range. A low O value means that the operation condition is distributed closely around the setpoint. A high O will result in a broader distribution around the setpoint.Use the output of the previous unit operation as input for the following one (Section 2.5). We assume that the starting data input for each unit operation is the output of the previous one [23]. The unit operation and its PP are connected based on the assumed process format.

### 2.1. Format of Risk Assessment

To conduct a comprehensive risk assessment, as suggested in the ICH Q9 guideline [3], each PP of each unit operation within the process has to be individually assessed for each CQA. The evaluation is based on the potential impact of each PP onto each CQA. The herein used RA includes the judged range, the set point of the PP as well as the estimated S value and the likely O value. S and O can range from one to five. For our study, it is assumed that the setpoint is in the center of the judged range of each PP. We evaluate each CQA at each unit operation considered within the process. Table 1 shows an example of the used RA.

### 2.2. Critical Delta CQA

Within Table 2, the last column refers to the value “Critical ∆CQA (%)." The herein entered value relates to the maximum acceptable loss of the CQA at a specific unit operation before OOS is reached. Each process has specific acceptance criteria at each unit operation and CQA. Such boundaries indicate that the process is able to meet DS specification or not. Since this value is mandatory for the linearization approach mentioned in Section 2.3, this CQA acceptance criteria needs to be calculated before doing the simulation. 

### 2.3. Severity Linearization

We assume that the process expert evaluates S to be ordinal scaled with equal distance to each level of the scale. Furthermore, we assume that the expert estimated S value based on its effect on the loss in CQA (Section 2.2), depending on the CQA. We assume that the process expert evaluates the setpoint of the judged range as the best operating condition. It does not matter if the PP operation point is on the left or right side of the setpoint, we assume an equal impact onto the CQA. It is further assumed that for this process the CQA is available as its maximum value, 100% in our case, after the first unit operation conducted within the entire process, which is generally the fermentation. To clarify these assumptions, we consider a critical loss in CQA (compare Section 2.2) to reach OOS of the first unit operation to be 10%, where the harmful effect is then lower than 90%. This critical loss can be considered as the critical drop in product concentration or critical lack of impurity removal evaluated per unit operation. Consider Figure 2, the y-axis shows the percentage loss of CQA while the x-axis presents the judged range of one example PP ranging from six to eight. At the center point of the judged range (seven in the specific case of Figure 2), we assume the maximum CQA value of 100%; while at both corner points (six and eight) the CQA value equals 100% minus the estimated critical ∆CQA (in this specific example leading to 90%). If we assume that the process parameter will have an equally harmful impact on CQA in both directions of the center point, we can assume an equally negative slope in both directions. However, without loss of applicability of the method, a harmful effect can also be assumed to occur only in one direction of the center point:(1)$$critical\;slope=\;\frac{\Delta Y}{\Delta X}=\frac{\Delta CQA}{\left(\frac{judged\;range}{2}\right)}$$

According to Equation (1), we calculate the critical slope. This value refers to the maximum S estimation. Since each PP has an S estimation, the current severity for each PP is calculated as:(2)$$current\;severity=\;\frac{critical\;slope}{\;maximum\;severity}\times estimated\;severity$$

Where the maximum severity is the highest S score within the RA. This procedure must be conducted for each PP evaluated within the RA.

### 2.4. Occurrence Assumption

We interpret the estimated occurrence value as the probability where within the judged range the PP can potentially vary. Based on the assumption mentioned in Section 2.3, that the setpoint is the optimal operation condition of the PP, the O value is crucial for estimating the probability distribution of the operation condition within the judged range of each PP. Considering Figure 3, we see two normal distributed distributions within the judged range of a PP, exemplary from six to eight. Histogram of Figure 3a represents the probability of the lowest occurrence value and indicates that the PP operation condition is close distributed around the setpoint. Histogram of Figure 3b, on the other hand, represents the probability at the highest occurrence value and indicates that the PP value can be at any condition equally often within the judged range. The width of the distribution is the standard deviation (std) and calculated as:(3)$$std=\;\frac{current\;occurrence}{\max occurrence}\times \;\frac{judged\;range}{2}$$

The remaining properties for distribution estimation, according the used python package, are the mean, and the lower and upper boundaries of the distribution, which are taken from the RA, as explained in Table 1. For estimating that PP operation condition within the distribution, we use the python package scipy Version 1.1.0 [24] especially the function stats.truncnorm, which results in a truncated distribution including just values within the judged range.

### 2.5. Integrated Process Modeling (IPM)

The integrated process aims to consider the entire process, where all assessed unit operations are connected consecutively. Zahel et al. [23] described the applied method and used a Monte Carlo [22] approach to include the error propagation within the unit operation concatenation. For our approach, we use just the knowledge from the conducted RA, and we never use experimental data. The following assumptions need to be considered:The mean of the output distribution of the previous unit operation is used as the starting point of the next unit operation.CQA acceptance criteria is used as a lower boundary of the respective unit operation.The severity of each PP indicates the slope of the CQA decreasing effect (see Section 2.3).The occurrence of each PP results in the distribution within the PP’s judged range where the value is randomly estimated (see Section 2.4).

Figure 4 shows the conducted integrated process model workflow, with a process including three unit operations. The black solid slope represents the critical slope (Equation (1)) from an individual PP while the red dashed slope indicates the corresponding slope of the PP considering the S assessment within the RA (Equation (2)). The distribution on the x-axis represents the distribution of the PP values where the PP can be measured. A sharp distribution indicates a low and a broad distribution indicates a high O estimation for the PP within the RA. The black dot-dashed arrow represents the mean of the output CQA distribution, which is assumed as the starting CQA value of the following unit operation, indicated by the green line. 

The concatenation workflow is shown in Figure 4, using the linearization of the S and the operation condition estimation based on the O value, the process can be described mathematically as: (4a)$$y\left(uo_{n}\right)=y\left(uo_{n-1}\right)+\overset{i}{\underset{i=1}{\sum }}(x_{i}-\;{\bar{x}}_{i\;judged\;range})\times \beta_{i_{upper}}\;\mathrm{for}\;x_{i}>\;{\bar{x}}_{i\;judged\;range}$$
(4b)$$y\left(uo_{n}\right)=y\left(uo_{n-1}\right)+\overset{i}{\underset{i=1}{\sum }}(x_{i}-\;{\bar{x}}_{i\;judged\;range})\times \beta_{i_{lower}}\;\mathrm{for}\;x_{i}<\;{\bar{x}}_{i\;judged\;range}$$
(4c)$$y\left(uo_{n}\right)=y\left(uo_{n-1}\right)-\overset{i}{\underset{i=1}{\sum }}|(x_{i}-\;{\bar{x}}_{i\;judged\;range})|\times \left|\beta_{i}\right|$$

For the specific case of equal slopes Equations (4a) and (4b) reduce to Equation (4c); compare Figure 1b2, where $y\left(uo_{n}\right)$ is the CQA value after simulating the current unit operation *n*, the intercept $y\left(uo_{n-1}\right)$ is the CQA of the previous unit operation n–1, $x_{i}$ is the estimated operation condition for the *i*^th^ PP (see Section 2.4), ${\bar{x}}_{i\;judged\;range}$ is the mean of the judged range (setpoint) of the *i*^th^ PP and $\left|\beta_{i}\right|$ is the absolute slope of the *i*^th^ PP (see Section 2.3). In case of unequal slopes, Equations (4a) and (4b) become valid, for different right or left slope, respectively. $\beta_{i_{upper}}$ and $\beta_{i_{lower}}$ represent the higher or lower slope value of a non-uniform judged range and become valid if $x_{i}>\;{\bar{x}}_{i\;judged\;range}$ or $x_{i}<\;{\bar{x}}_{i\;judged\;range}$, respectively. The holistic numerical simulation of all unit operations until DS, evaluated within the RA, is considered as one cycle. To ensure reproducibility of the IPM, a Monte Carlo is conducted where 1000 cycles were simulated [22,23]. 

For OOS event assessment, the mean value of the CQA at DS is calculated, after 1000 simulated cycles, including all assessed PPs. In the next step, we randomly removed one PP individually, repeated the Monte Carlo approach, and finally calculated the mean of CQA at the DS. The excluded PP was added and this procedure was repeated until every existing PP was removed once. Afterward, we calculated the OOS events of the DS CQA distribution between each simulation where one PP was removed and compared with the simulation containing all PPs ($OOS_{total}$). This results in the OOS reduction if we exclude one PP from the process (Equation (5)).
(5)$$OOS\;reduction_{PP}=\;OOS_{total}-OOS_{without\;PP}$$

Based on a limit estimated by the process expert, the CPPs can be identified by evaluating the PP’s OOS reduction. A PP with low OOS reduction is not potentially critical effecting the process, while a PP with a high OOS reduction potentially affects the process (see Figure 1b2).

### 2.6. Case Study

A risk assessment was performed to determine the process parameters that would affect the yield of a biopharmaceutical process. The herein considered RA includes the following unit operations.

In the case study the classic RPN method was followed for selection of PP to study in a DoE. The data from the RA using the RPN method will be compared to data obtained from the IPM method. 

### 2.7. Software

Within this study, the RA was conducted within MS Excel 2016 (Microsoft, Redmond, WA, USA), the algorithm was developed and the simulation was conducted within Python 3.5.4 (Python Software Foundation) [25].

## 3. Results

The proposed method to evaluate PPs and their effect on CQA based on RA was carried out in-silico to verify the algorithm. Next a case study was presented where the method was applied to data obtained from an RA performed for a vaccine production process. 

For the in-silico study, we assumed a maximum S and O value of five. We aimed to identify the contribution of S and O values to the potential loss in CQA (compare Section 2.2). To ensure a comprehensive algorithm assessment, 25 PPs were simulated using each possible S and O combination. Therefore, we assume that the judged range of the PP itself has no impact on the reduction event. Hence, the judged range of the simulated PP was identical between the simulations. We consider a starting CQA concentration of 100% and two unit operations for this study.

For the case study, an FMEA was performed for a vaccine production process; an overview is given in Table 2. We assume a starting CQA value of 100%. The analysis is focused on two CQAs: product yield and as impurity host cell protein (HCP) concentration. For algorithm testing, the simulation was conducted for both CQA cases. We show that the developed algorithm can be used for one which aims for a high CQA concentration (product yield) and one seeking a low CQA concentration (HCP) at DS. 

### 3.1. In-Silico Simulation

In the first step, we simulated two unit operations with 10 and 15 PPs, respectively. This resulted in 25 PPs where any possible combination of S and O occurred once. To simulate the S and O influence on the process, we considered the IPM technology using the Monte Carlo approach discussed in Section 2.5. Figure 5 shows an ascending sorting of the CQA reduction per PP. The y-axis represents the reduction in CQA (%) value and the x-axis the PP. It can be seen that the PP with the smallest S and O has the lowest CQA reduction while the PP with the highest S and O has the highest one, which was expected. 

However, by considering the PPs individually, a different impact of the S and O value to the CQA reduction was observed. The individual influence in the decrease from S and O is displayed in Figure 6. While Figure 5 represents a quantitative decision tool to evaluate the probability of the OOS events individually, the weighted influence of the S and O is shown in Figure 6. The presented contour plot shows the difference in the weight on the CQA reduction from S and O separately. The figure shows that the S value has a higher impact, since CQA reduction increases faster, than the O value, and therefore a higher criticality for the entire process. 

To evaluate the difference of the IPM simulation to the standard RPN approach, we plotted the simulated OOS reduction values against their corresponding RPN value. Figure 7 shows the result. We calculated for each of the 25 PPs used within the in-silico study the RPN value like:(6)RPN=Severity×Occurrence

Equation (6), only considers the product of S and O as RPN since we compare the S and O estimation from RA interpretation with the impact based on the linearization approach. If we follow the RPN assessment approach, we would assume a positive linear correlation between the RPN value and contribution to CQA reduction. In Figure 7, we see that, based on our model, this assumption is not valid as a saturation curvature is observed. In the figure, we plotted the resulting CQA reduction of the PP on the y-axis against the corresponding RPN value on the x-axis. It is clearly shown that at an RPN value of 20 two PPs are present, both with a clear different impact on process performance (here measured in CQA reduction). Due to the already presented different contribution of S and O to event reduction, this relation behavior was expected and underlines that the RPN assessment approach is not the ideal solution for evaluating the criticality of a PP.

As shown in Figure 7, the common RA assessment strategy based on the RPN approach results in misinterpretation of the criticality of a PP. If we study the two PPs, within Figure 7, with an RPN value of 20 more closely, it can be seen that one of them has an S of four and an O of five (Case 1) while the other one has an S of five and an O of four (Case 2). If we consider a CQA reduction of ≥10% as critical for the process, we would identify Case 1 as a critical PP. It can be seen that Case 1 reduces the CQA by ~12% while for Case 2, the reduction is just ~8%. Cases 3 and 4 should be considered as potentially critical. Case 3 is the one with the maximum S and O, RPN equals 25 and CQA reduction of ~12%. Case 4 is the PP with an S of five and an O of three and CQA reduction of ~11%. 

While for the RPN approach, we would set an arbitrary boundary that every PP equal to or higher than 20 PRN is considered as potentially critical. If so, we would include the PPs from Case 1, 2, and 3 and never identify the PP from Case 4, although it has a higher impact on the CQA reduction than the PP from Case 2. It is shown that based on the different contribution of the S and O factors to the CQA reduction, we do not recommend using the product of the S and O to assess the criticality of a PP based on a holistic threshold. 

### 3.2. Case Study

To evaluate the simulation based on a real RA, we considered a risk assessment carried out for a vaccine production process provided by a leading research and development (R&D) institute for vaccine research. The process unit operations are given in Table 2. We prove the simulation on the key performance indicators product yield (further referred as yield) and on the HCP impurity. 

#### 3.2.1. Yield Simulation

The result of the simulation by using the yield as CQA is shown in Figure 8. The plot shows the % CQA reduction per PP, with the name of the PP being a combination of the unit operation (see Table 2) and the corresponding PP name. The red marked PPs represent the parameters the expert considered as potentially critical based on their standard RPN evaluation approach.

We see that if we consider the conducted simulation and assume a PP to be critical if CQA reduction is at least 10%, just one PP would be considered as relevant to be evaluated in a DoE instead of seven as suggested by the process experts. This reduction in DoE factors, also shown using a different method by Borman et al. [26], would result in an increase power of the DoE and a decrease in experimental effort.

#### 3.2.2. HCP Simulation 

A further case study was conducted showing the simulation application for impurity assessment. We considered the HCP as CQA and did the IPM as discussed in Section 2.5. The result is shown in Figure 9. We see that the maximum CQA reduction is 6% and that the PP contribution to the response varies compared with the PP in Figure 8. We show that results of the yield and the HCP simulation are not identical, and different factors influence different CQAs. This result enforces the need of the individual assessment for each CQA. It is strongly recommended to evaluate the reduction for each CQA to consider the most contributing PP. We show that any CQA can be simulated with the developed approach. It does not matter if the CQA is a key performance indicator linked to product yield or an impurity; the simulation is applicable for both types.

## 4. Discussion

We propose a new RA evaluation approach which avoids the limitation of using RPNs and relates the risk quantitatively to the probability of drug substance CQAs being out of specification. We demonstrate a tool which evaluates the process by considering all unit operations until DS, leading to a holistic process evaluation instead of a single unit operation assessment. We show that the above-conducted simulation is based purely on estimations from the process experts. Only the evaluation within the RA is the basis for the simulation where the factors S and O are the most crucial ones. This section explains the above-shown results in detail and discusses the improvements of the developed algorithms compared with the RPN approach.

### 4.1. CPP Assessment 

Within this study, we demonstrate an approach that allows us to make a rational and quantitative assessment of the criticality of a PP based on its contribution to CQA reduction. A particular threshold identifies the PP as critical for the process. This threshold has to be verified by the process expert for each CQA. The assessment is based on the CQA reduction, which is generated by iterative simulation while always one PP is removed from the simulation. Such a result is shown in Figure 5 and needs to be verified for each CQA individually. This result shows the need to consider DoEs for each CQA separately by including the critical factors shown in Figure 5. 

#### 4.1.1. Assessment Interpretation

For interpretation of Figure 5, the upper and lower specification limits of CQAs in the DS have to be known. These limits are typically generated by:Using historical process dataIn the case of a new process, conducting one setpoint process and use the occurred CQA value as a reference value to estimate the specifications for CQA at DSUse literature data or pharmacopeia to set the specification limits for CQA at DS

In our example, we considered everything as OOS if the CQA value at DS was reduced by at least 10%. 

However, it is necessary to interpret the simulation of each CQA independently as it is not recommended to use a general criticality threshold for the process which indicates that a PP is potentially critical or not. For each CQA simulation, a different contour plot, as shown in Figure 6, occurs and needs to be interpreted based on the prior defined specification limits and related OOS events. This plot allows us to:get a more profound process understandingidentify the weighting of S and O to each CQAidentify potentially critical PP based on a rational decision-making tool.

#### 4.1.2. Integration of Factor D (Detectability)

As usual for an FMEA, factor D is evaluated for each PP to each CQA. This factor represents the ability of detecting the failure and the timely detection of the failure (consider the duration for conducting a measurement). We assumed in the above in-silico study that the measurement results are always immediately available and equally critical to the process. If that is not the case, the simulation has to be conducted as usual and the CQA reduction assessed for each PP and CQA combination. When D is taken into account, PPs are grouped into four to six blocks, containing the same amount of PPs in order by their contribution to CQA reduction. Four blocks are selected when the amount of PPs is lower than 28. When more PPs are considered six blocks are chosen. In the next step, per block, PPs are sorted by their D factor. This sorting results in additional weighing of the PPs. The therein highest PP is considered as most critical within each block. It is possible that the CQA reduction is not the highest while the PP is considered as most critical. When this is observed, all PPs of that certain block should be considered as critical until no PP is found to be the critical regarding CQA reduction. 

Considering Figure 8, four blocks would be present, if D evaluation was available. Block 1, 2, and 3, each consist of six PPs and start with PP, Con1_Var1, Conc1_Var2, and Conc1_Var4 respectively. Block 4 consists of five PPs starting with the PP Conc2_Var4. 

### 4.2. Comparing RPN vs. IPM Approach

As discussed in Section 1, several disadvantages of the RPN approach are present [7,8]. Within this study, we presented a methodology to avoid all of the described pitfalls. 

#### 4.2.1. Multiplication of Ordinal Scaled Values

As a result of the used IPM approach, the multiplication step of S and O is not further required. For the simulation, we translate the S and O values to meaningful information used for process interpretation.

#### 4.2.2. Gaps within the RPN Scale

The reason for the gaps within the RPN scale is due to the conducted multiplication step, which is not further done for the herein presented approach. The gaps present in the CQA reduction are expected and acceptable for this kind of analysis. 

#### 4.2.3. Different Meaning Although RPN is Equal

Within Figure 6, it is shown that the severity and occurrence have a different effect on the CQA reduction. This observation shows the importance of a reasonable PP assessment for each factor individually. We cannot generally assume that an O estimation of five always results in a critical process parameter while the S is two, or vice versa. We have to individually assess each PP by its CQA reduction effect and not if just one factor is at maximum.

#### 4.2.4. Big Influence by Small Changes

Consider Figure 5. It is shown that small changes also result in a minor increase in CQA reduction. The height of the bar indicates this. It is also demonstrated that influence also depends on the factor. Therefore, such an impact cannot be generalized, that means an element with a more significant weight than the other also results in a stronger influence proportional to the other one.

#### 4.2.5. All Assessed Factors are Weighted Equally

As discussed in Figure 6, it is clearly shown for this study that S and O are not equally contributing to the OOS. The sharp increase of the S value shows a more prominent influence of S than O. We therefore recommend being careful by estimating the S and O values for each PP and do not solely assess by feeling. The values are best supported by data or arguments.

## 5. Conclusions

This study aimed to present a new RA evaluation approach based on objective quantification of each PP. This assessment approach can be used instead of the usual RPN evaluation approach and is based on calculating the critical influences of PP. We conducted an IPM to estimate the OOS events at DS instead of an arbitrary RPN threshold. Every standard RA format, where S and O are estimated as severity on a CQA for occurrence (or probability) for a PP, can be used for this simulation. 

Finally, we show that each CQA has to be assessed individually and that according to our interpretation for S and O, their contribution to the PP criticality is unequal. This observation leads to the conclusion that the typical RA generation has to be done with more diligence than before relating to S and O factor estimation.

To implement that approach, the RA needs to follow the prerequisites mentioned in Table 1 and the additional critical loss in CQA for each unit operation needs to be assessed. The simulation can be conducted within python or any other programming language without the use of any special libraries. The final contribution to the CQA reduction shows a quantitative impact on the CQA and allows a more reasonable PP critical assessment than the prior calculated RPN. Already one unit of operation and two process parameters are sufficient to simulate the CQA reduction. 

## Figures and Tables

**Figure 1 bioengineering-06-00114-f001:**
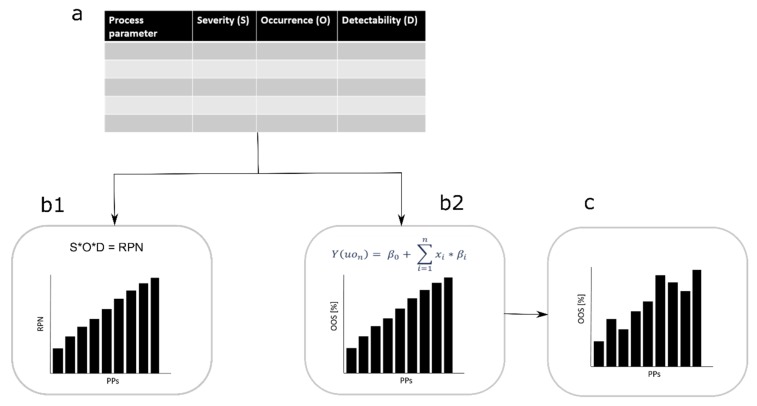
Workflow comparison, classical failure mode and effects analysis (FMEA) to new approach. (**a**) Risk assessment is conducted and severity (S), occurrence (O), and detectability (D) of each process parameter (PP) for each critical quality attribute (CQA) is evaluated. By considering the standard approach, the three factors are multiplied to generate the risk priority number (RPN) number which ranks the PP from the lowest to the highest RPN value (**b1**). In our approach a linearization of the S and O evaluation is done and a simulation is conducted. We consider the factor S as the model size (beta value in the linearization equation) and the factor O as the parameter distribution (x value in the linearization equation). Further, the out of specification events can be estimated and ranked by their appearance from the lowest to highest (**b2**). (**c**) Finally, the factor D influences the importance of the PP criticality on the CQA.

**Figure 2 bioengineering-06-00114-f002:**
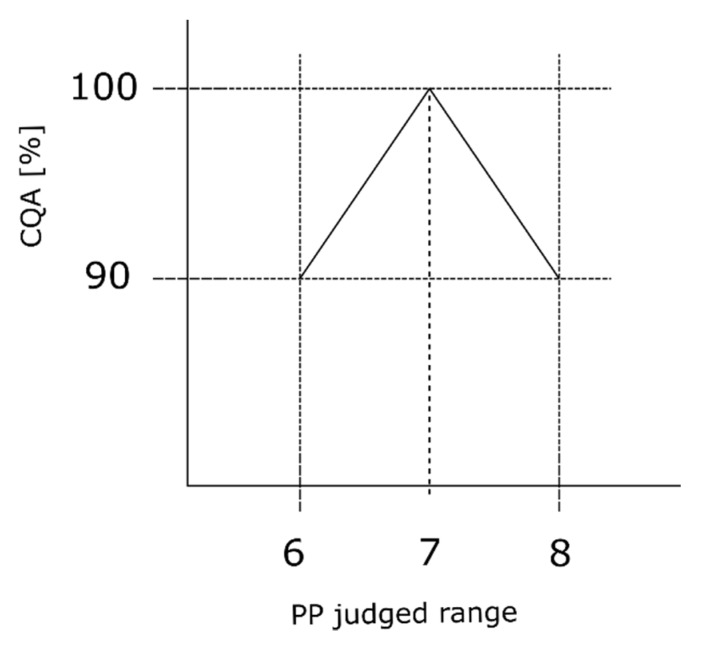
Severity interpretation reflecting the process expert’s assessment. The observed triangle shape reflects that assumption when estimating the severity value. We assume that at setpoint, the center point of the judged range, the smallest effect on the CQA is present. While the further away from the setpoint and the closer to the border of the judged range, the effect onto the CQA increases. The herein calculated slope represents the effect of the maximum severity value within the RA and has to be adjusted for each PP based on its respective severity value.

**Figure 3 bioengineering-06-00114-f003:**
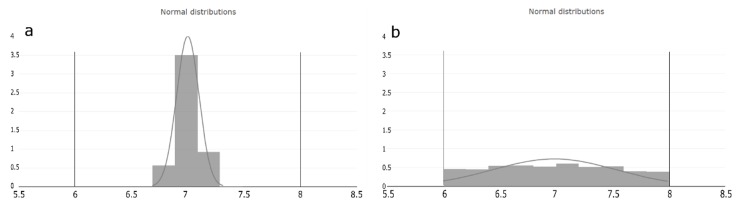
Occurrence value interpretation based on the process expert assumption. The horizontal lines on the x-axis represents the judged range of the PP and the y-axis probability shows the estimated distribution area of the (**a**) lowest and (**b**) highest occurrence value. (**a**) represents the distribution of the lowest possible occurrence value. It can be seen that the probability of the PP operation condition is very close to the setpoint of the judged range. (**b**) represents the distribution of the highest possible occurrence value. It can be seen that the probability that any operation condition within the judged range can occur is equal.

**Figure 4 bioengineering-06-00114-f004:**
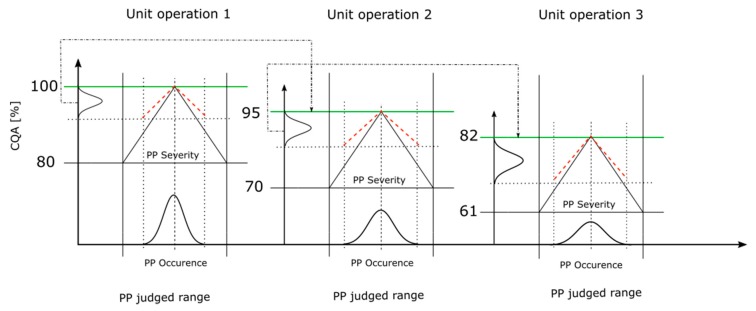
Integrated process model workflow used to concatenate all the assessed unit operations from the risk assessment (RA). The y-axis represents the CQA ranges where the CQA distribution is located and the x-axis represents the judged range of the PP. Based on the S and O value assessment for each PP, the CQA distribution is determined somewhere within the CQA range. The upper CQA range is the mean of CQA from the predecessor unit operation and the lower CQA range is the critical CQA value per unit operation. The black slope represents the critical effect of the PP. The red dashed line represents the corresponding slope of the PP’s S assessment. The distribution on the x-axis and within the PP’s judged range indicates a low, sharp distribution (as in UO 1), or large, broad distribution (as in UO 3), of the O assessment. The black dot-dashed arrow shows the transfer from the mean of the CQA distribution as the upper boundary of the next CQA range, indicated by the green line.

**Figure 5 bioengineering-06-00114-f005:**
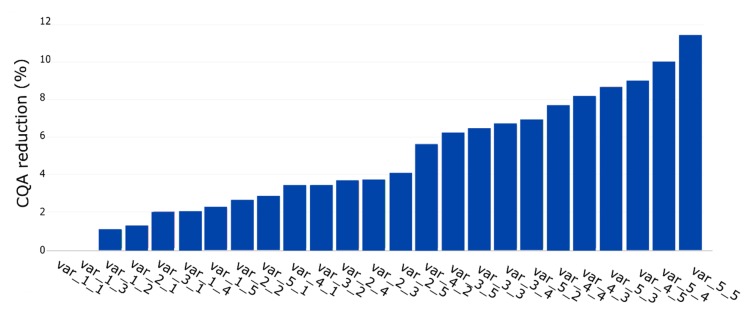
Quantitative assessment of each PP ascending ranked by its contribution to CQA reduction (%). First and second values on each name correspond to S and O, respectively.

**Figure 6 bioengineering-06-00114-f006:**
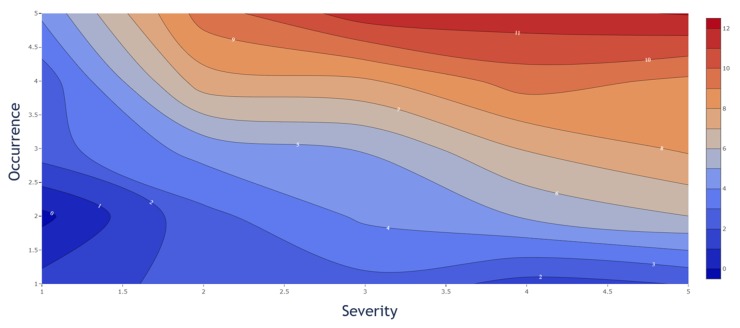
Severity and occurrence contribution to CQA reduction (%). The severity shows a more significant impact as is illustrated by an earlier increase in % CQA reduction.

**Figure 7 bioengineering-06-00114-f007:**
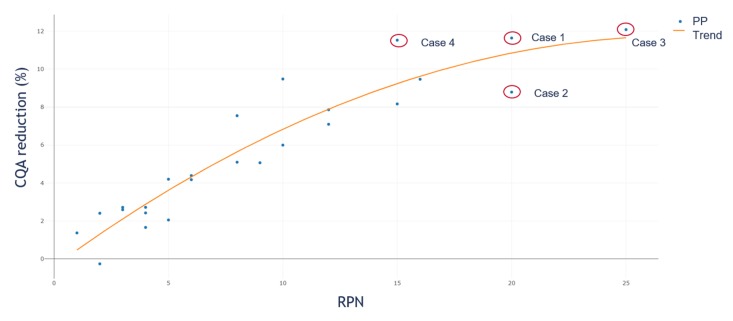
CQA reduction against corresponding RPN. This saturation curvature shows that we have to reject the assumption of the RPN assessment approach, where we assume a linear increase in PP criticality as high as the RPN value. Such behavior concludes that we have to assume different weights for severity and occurrence.

**Figure 8 bioengineering-06-00114-f008:**
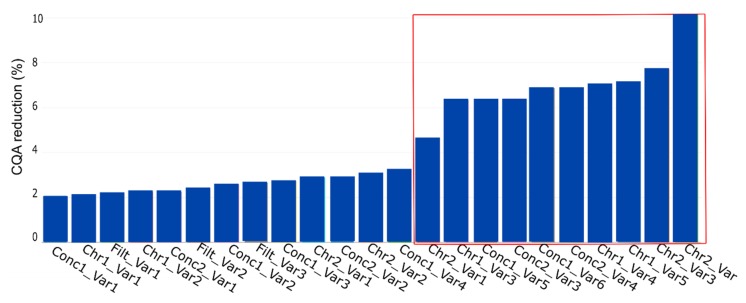
PP influence on CQA reduction (%) for the key performance indicator yield. This plot represents the contribution of each PP, contributing to CQA reduction at the drug substance (DS). The PP name is the combination of the unit operation abbreviation (Table 2) and the parameter of interest. The plot shows the contribution in ascending order, which indicates the highest impacting PP on the rightest side. The variables inside the red lines indicate the potential critical PP as assessed by the process experts.

**Figure 9 bioengineering-06-00114-f009:**
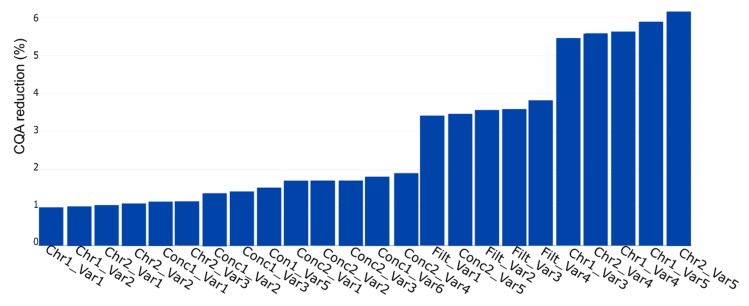
PP influence on CQA reduction (%) for host cell protein (HCP) concentration. The bar plot represents the contribution of each PP to possible out of specification (OOS) events at drug substance. The plot shows the contribution in ascending order, which indicates the highest impacting PP on the rightest side.

**Table 1 bioengineering-06-00114-t001:** Suggested FMEA format with example values. UO represents each unit operation within the process. PP is the name of the process parameter. Judged range lower and upper represents the suggested ranges of the PP within the assessment, respectively. The setpoint is that of the PP within the evaluation. CQA 1 is the quality attribute where the PP impact is estimated. Severity is the experts estimated S value of the PP to the corresponding CQA. Occurrence is the experts estimated O value of the PP to the corresponding CQA. Detectability is the experts estimated D value of the PP to the corresponding CQA. Critical ∆CQA (%) is the significant percentage loss in CQA at a particular unit operation (introduced in Section 2.2).

					CQA1			
UO	PP	Judged Range Lower	Judged Range Upper	Setpoint	Severity	Occurrence	Detectability	Critical ∆CQA (%)
**UO-1**	PP1	6	8	7	4	3	2	
	PP2	30	70	50	3	4	2	
								10
**UO-2**	PP1	6.8	7.6	7.2	5	2	2	
	PP2	32	38	35	2	4	4	
								10

**Table 2 bioengineering-06-00114-t002:** Overview of assessed unit operation within the RA.

Consecutive Order of the Unit Operation	Unit Operation	Unit Operation Abbreviation	Scope for the Simulation	Number of Assessed PPs	Number of Assessed CQAs
1	Fermentation	Ferm	No	53	27
2	Filtration	Filt	Yes	5	28
3	Concentration 1	Conc1	Yes	7	24
4	Concentration 2	Conc2	Yes	7	24
5	Chromatography 1	Chr1	Yes	8	28
6	Chromatography 2	Chr2	Yes	7	28
7	Formulation	form	No	4	28

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
