# Peer review of "Quantitative CPP Evaluation from Risk Assessment Using Integrated Process Modeling"

_bioengineering, 2019, doi:10.3390/bioengineering6040114_

Round 1

Reviewer 1 Report

The paper deals with the quantitative evaluation of the critical process parameters of a biopharmaceutical production process through risk assessment using an integrated process modelling approach. The paper is clear and well written. Few comments are reported in the following:

Page 1, Abstract: “This algorithm quantitatively describes the OOS probability for each PP assessed within the RA”. In my opinion, the term “specifications” is not fully appropriate when referred to the PPs. It is more related to the expected value that the manufacturer desires for its product (and accordingly related to the CQAs). The PPs are subject to (operating) constraints. Some problems that are present in the whole manuscript:
a) two figure 1 are present, so the numeration of the figures is incorrect;
b) I guess that the conversion to the pdf version created some problems with the number of sections and some cross references (see lines 121, 126, 131, 152, 156, 223, etc..);
c) the figure captions should explain what is shown in the figure, the method descriptions and the results should be discussed in the main text. Page 3, first Figure 1 (line 89): the explanation of the methodology is not completely clear. I think that the authors should improve it. Bullet list at the end of Page 3 and the beginning of page 4: is the procedure mainly oriented to the univariate consideration of PPs and CQAs? What if data are highly correlated? Figure 2: these figures display a very high and a very low occurrence value, not the highest, nor the lowest. The highest corresponds to the uniform distribution where the standard deviation is infinite, the lowest the one with zero standard deviation. Page 9: I do not think that the following sentence (referred to Figure 5) is correct: “The presented contour plot shows the difference in the weight on the CQA reduction from S and O separately. The figure shows that the S value has a higher impact, since CQA reduction increases faster, than the O value and therefore a higher criticality for the entire process.”. From Figure 5 it can be seen that for medium and high values of S, if S is held constant all the range of % CQA reduction can be found depending on the O value. Instead, the variability of the % CQA reduction for a constant value of O seems to be much more limited. Page 11, lines 350-351: what research institute? What vaccine? Could not the authors mention them? Page 12, line 397: what is the impact of considering CQAs individually? Is it possible to consider them jointly? How? Page 14, lines 452-453: why reasoning only in terms of CQA reduction? Could not the author reason in terms of deviation from the target (both increase or decrease of the value of the critical quality attribute from the target)? The authors miss commenting the limitations of the proposed methodologies.

Author Response

Response to Reviewer 1 Comments

Pont1:

Page 1, Abstract: “This algorithm quantitatively describes the OOS probability for each PP assessed within the RA”. In my opinion, the term “specifications” is not fully appropriate when referred to the PPs. It is more related to the expected value that the manufacturer desires for its product (and accordingly related to the CQAs). The PPs are subject to (operating) constraints.

Answer: Thank you for that comment. We adjusted the mentioned sentence in the abstract. Please consider changes in line 22.

Some problems that are present in the whole manuscript:

Point2:

a) two figure 1 are present, so the numeration of the figures is incorrect;

Answer: Thank you for that comment. We checked and adjusted the Figure caption in the manuscript

Point3:

b) I guess that the conversion to the pdf version created some problems with the number of sections and some cross references (see lines 121, 126, 131, 152, 156, 223, etc..);

Answer: Thank you for that comment. We check the manuscript and updated the cross reference on the necessary parts.

Point4:

c) the figure captions should explain what is shown in the figure, the method descriptions and the results should be discussed in the main text. Page 3, first Figure 1 (line 89): the explanation of the methodology is not completely clear. I think that the authors should improve it.

Answer: Thank you for that comment. We adjusted the figure caption according your suggestion.

Point5:

Bullet list at the end of Page 3 and the beginning of page 4: is the procedure mainly oriented to the univariate consideration of PPs and CQAs? What if data are highly correlated?

Answer: Thank you for that comment. Yes, based on the Severity assumption we can mainly resolve main effect of PP on CQA.

Point6:

Figure 2: these figures display a very high and a very low occurrence value, not the highest, nor the lowest. The highest corresponds to the uniform distribution where the standard deviation is infinite, the lowest the one with zero standard deviation.

Answer: Thank you for the comment. We partly agree with that comment. Figure 3a shows the sampling distribution if the Occurrence of a PP is estimated to 1, therefore, the lowest possible occurrence value, since it not commonly used to have an Occurrence of 0. If that is the case the std is zero. Figure 3b shows the sampling distribution if the highest possible Occurrence value is assumed. Since we calculate the std from equation 3 the mentioned statement is correct since we cannot assume an infinite std, since we have to consider the judged range of the PP. Although, from a statistical point of view, your mentioned statement is correct we have to consider the judged range within the std calculation, therefore, no chances in the manuscript were made.

Point7:

Page 9: I do not think that the following sentence (referred to Figure 5) is correct: “The presented contour plot shows the difference in the weight on the CQA reduction from S and O separately. The figure shows that the S value has a higher impact, since CQA reduction increases faster, than the O value and therefore a higher criticality for the entire process.”. From Figure 5 it can be seen that for medium and high values of S, if S is held constant all the range of % CQA reduction can be found depending on the O value. Instead, the variability of the % CQA reduction for a constant value of O seems to be much more limited.

Answer: Thank for that comment. We partly agree with that comment. Your explanation is true, therefore we assume a higher impact of the S value since the % CQA reduction reaches the critical level faster than the O value. If O is held constant on a certain medium level, we can assume any S and the PP is not considered as critical, O has therefore less impact on the CQA. If we keep S constant on a certain medium level, we will assume a critical PP if the O increases to the maximum level, that shows that the S has more impact on the CQA since medium S values impact more than medium O values.

Point8:

Page 11, lines 350-351: what research institute? What vaccine? Could not the authors mention them?

Answer: Thank you for that comment. The research institute can be found in the affiliation of the co-authors. The investigate vaccine cannot be mentioned yet, since disclosed due to confidential issue and out of scope of that manuscript. Therefore, we didn’t add any further information in the suggested lines.

Point9:

Page 12, line 397: what is the impact of considering CQAs individually? Is it possible to consider them jointly? How?

Answer: Thank you for that comment. We partly agree with your comment. We often considered the problem in a DoE evaluation that factors are selected based on one CQA assessment. The result of the conducted experiments are further used to identify effects on multiple CQAs although just the critical PP from one CQA are included in the DoE. This make the effect assessment for all the other CQA useless and not meaningful. We want to state with that sentence, that you have to consider each CQA individually in the DoE planning, which based on the results from the RA. We added a sentence for clarification at the end of section 4.1.

Point10:

Page 14, lines 452-453: why reasoning only in terms of CQA reduction? Could not the author reason in terms of deviation from the target (both increase or decrease of the value of the critical quality attribute from the target)? The authors miss commenting the limitations of the proposed methodologies.

Answer: Thank you for that comment. We observed that each PP has a particular contribution to the CQA distribution after the last unit operation. We simulate the process with each PP included and got an estimated CQA distribution after the last Unit operation, where we calculated the mean. Then we exclude one PP and repeat the simulation, again the CQA distribution after the last unit operation is estimated and the mean is extract. The new mean is always lower than mean, where all PP are included. This difference can be considered as the PP contribution to the CQA. This verifies our assumption from Figure 2, that if you run an experiment far away from the set point you reduce your CQA. Therefore, we only mention a reduction in CQA since we only calculate a loss and never an increase.

Reviewer 2 Report

The approach described is worthy of publication as qualitative or at best semi quantitative approaches to performing risk assessments are currently common practice in the pharmaceutical industry. The approach discussed better weights the contributions from Severity and Occurrence. It would helpful if the authors could quickly summarise how this approach could be implemented and clearly articulate the prerequisites in the conclusion section.

General points

In the abstract it correctly states that RAs identify links between PPs and quality attributes. At the start of the introduction section it states that RA evaluates risks relating to failing DS specification. It should be noted that there are sometimes quality attributes which are on the specification which are not critical to quality (e.g. appearance of drug substance). Maybe the authors should explicitly mention that RAs need to consider risks to failing CQA specs as well as other attributes which may have to be specified.

Line 119/161/284 etc: Need to explain in more detail what you mean by ‘critical loss of CQA’ as this concept is key to the paper. Also on line 290 what is meant CQA value of 100% ? This would make sense for purity but not for other CQAs (e.g. impurities)

Line 320/321: You have previously defined RPN (in Figure 1) as SxOxD. However here you only state SxO – this needs explaining as you’re only considering part of the equation

Line 367/368: You can also reduce the power of the DoE by grouping together low risk factors or by performing saturated designs – suggest you reference the following publication as another way of decreasong experimental effort in relation to DoEs:

Reduced method robustness testing of analytical methods driven by a risk-based approach, P Borman, M Chatfield, P Jackson, A Laures, G Okafo, Pharmaceutical Technology Europe 4 (22), 2010

It would be advisable to add a section in the conclusion which explains briefly how easy it is in practice to apply this more quantitative data-driven approach versus traditional use of FMEA via RPN. What is the minimal amount of input/prior knowledge needed for each CQA before performing the risk assessment?

Detailed Comments:

Line 14: Process parameters are usually identified before performing formal risk assessments (e.g. via construction of parameter-attribute matrix). Therefore suggest Change to ‘Risk Assessments are frequently conducted to assess the potential effects of process parameters (PP) on product quality attributes.’

Line 15/16: Suggested rewording: ‘The evaluation of the criticality of PPs is based…’

Line 20: Suggested rewording: ‘To avoid these pitfalls and to assess the probability of drug substance CQAs being out of specification, we present…’

Line 34/35: Suggested rewording: ‘Risk Assessment (RA) is a prominent tool for evaluating the risks relating to not meeting drug substance specification, within a biopharmaceutical process

Line 41 : Suggest change factors to parameters

Line 44: invert order as severity should be identified first: ‘… severity of consequence and the probability of its occurrence.’

Line 46: change ‘pushing forward actively’ to ‘implemented’

Line 47/48: Change to ‘As a RA method, a Failure Mode and Effects Analysis (FMEA) tool aims to determine how a process can fail and evaluate the effect of this failure on the product [7,8]’

Line 59: Not sure I agree with this statement. S, O and D are usually scored on the same ordinal scale. E.g. 1-5 as in the paper?

Line 64 does not make sense. Do you mean ‘RPNs with the same score but derived differently?’

Line 126/127 – Change to ‘We assume the process expert evaluates the change of each PP within the judged range.

Line 131 – reference missing

Line 152 – reference missing

Line 182 – this should be figure 2 as figure 1 appears on line 90. Need to check text where you are referencing to figure 1 and figure 2

Line 200 – reference missing

Line 332 – change ‘reject’ to ‘rejected’

Line 384/385: We propose a new RA evaluation approach which avoids the limitations of using RPNs and relates the risk quantitatively to the probability of drug substance CQAs being out of specification

Line 434 – reference missing

Line 441 and 442 – Scala? Do you mean Scale?

Author Response

Response to Reviewer 2 Comments

Point1:

The approach described is worthy of publication as qualitative or at best semi quantitative approaches to performing risk assessments are currently common practice in the pharmaceutical industry. The approach discussed better weights the contributions from Severity and Occurrence. It would helpful if the authors could quickly summarise how this approach could be implemented and clearly articulate the prerequisites in the conclusion section.

Answer: Thank you for that comment. We added a section at the end of the conclusion which addresses that prerequisites and implementation.

General points

Point2:

In the abstract it correctly states that RAs identify links between PPs and quality attributes. At the start of the introduction section it states that RA evaluates risks relating to failing DS specification. It should be noted that there are sometimes quality attributes which are on the specification which are not critical to quality (e.g. appearance of drug substance). Maybe the authors should explicitly mention that RAs need to consider risks to failing CQA specs as well as other attributes which may have to be specified.

Answer: Thank you for that answer. We updated the sentence at page 2 line 48.

Point3:

Line 119/161/284 etc: Need to explain in more detail what you mean by ‘critical loss of CQA’ as this concept is key to the paper. Also on line 290 what is meant CQA value of 100% ? This would make sense for purity but not for other CQAs (e.g. impurities)

Answer: Thank you for that comment. Section 2.2 explains the term “critical loss in CQA” in a detailed overview. We added the cross reference to section 2.2 to each section where this is mentioned, compare line 163 and 286.

We partly agree with your second question. We just assume that the CQA is available at 100% (everything from the CQA) after the first unit operation. For impurity we have also 100 % available a particular amount of eg. HCP is available. Therefore, we can use this term holistically, since we can estimate the loss of the CQA [%] for both and need to evaluate the result based on purity or impurity.

Point4:

Line 320/321: You have previously defined RPN (in Figure 1) as SxOxD. However here you only state SxO – this needs explaining as you’re only considering part of the equation

Answer: Thank you for that comment. We added a sentence, consider line 323+324

Point5:

Line 367/368: You can also reduce the power of the DoE by grouping together low risk factors or by performing saturated designs – suggest you reference the following publication as another way of decreasong experimental effort in relation to DoEs:

Reduced method robustness testing of analytical methods driven by a risk-based approach, P Borman, M Chatfield, P Jackson, A Laures, G Okafo, Pharmaceutical Technology Europe 4 (22), 2010

Answer: Thank you for that comment. We adjusted the sentence and added the reference, see lines 370+371.

Point6:

It would be advisable to add a section in the conclusion which explains briefly how easy it is in practice to apply this more quantitative data-driven approach versus traditional use of FMEA via RPN. What is the minimal amount of input/prior knowledge needed for each CQA before performing the risk assessment?

Answer: Thank you for that comment. We added a section to the end of the conclusion.

Detailed Comments:

Point7:

Line 14: Process parameters are usually identified before performing formal risk assessments (e.g. via construction of parameter-attribute matrix). Therefore suggest Change to ‘Risk Assessments are frequently conducted to assess the potential effects of process parameters (PP) on product quality attributes.’

Answer: Thank you for that comment. We changed the suggested sentence accordingly, please compare line 14+15.

Point 8:

Line 15/16: Suggested rewording: ‘The evaluation of the criticality of PPs is based…’

Answer: Thank you for that comment. We changed the suggested sentence, please consider line 15+16.

Point9.

Line 20: Suggested rewording: ‘To avoid these pitfalls and to assess the probability of drug substance CQAs being out of specification, we present…’

Answer: Thank you for that comment. We removed the word pitfalls and replaced it with misinterpretation. Please consider line 20.

Point 10:

Line 34/35: Suggested rewording: ‘Risk Assessment (RA) is a prominent tool for evaluating the risks relating to not meeting drug substance specification, within a biopharmaceutical process

Answer: Thank you for that comment. We changed to sentence, please consider line 34+35

Point11:

Line 41 : Suggest change factors to parameters

Answer: Thank you for that comment. We changed the sentence accordingly, please consider line 41.

Point 12:

Line 44: invert order as severity should be identified first: ‘… severity of consequence and the probability of its occurrence.’

Answer: Thank you for that comment. We changed to order according your suggestion, please consider line 44.

Point 13:

Line 46: change ‘pushing forward actively’ to ‘implemented’

Answer: Thank you for that comment. We changed the sentence accordingly, please consider line 46.

Point 14:

Line 47/48: Change to ‘As a RA method, a Failure Mode and Effects Analysis (FMEA) tool aims to determine how a process can fail and evaluate the effect of this failure on the product [7,8]’

Answer: Thank you for that comment. We changed the suggested sentence accordingly, please consider Line 47+48.

Point15:

Line 59: Not sure I agree with this statement. S, O and D are usually scored on the same ordinal scale. E.g. 1-5 as in the paper?

Answer: Thank you for that comment. S, O and D are ordinal scale, since the values have a ranking of that corresponding value. Each number tells us how much of a property an item has, therefore, we need to consider S, O and D as ordinal scaled. If you consider reference 7 (Bowles, J.B. An assessment of RPN prioritization in a failure modes effects and criticality analysis. In Proceedings of the Reliability and Maintainability Symposium, 2003. Annual; IEEE, 2003; pp. 380–386.) the different of normal and ordinal scale is descript in details. The multiplication is possible but not meaningful. The only way you can multiply ordinal scaled values, if they have equal distance to each other, therefore, we explicitly mention in the manuscript, in section 2.3, that we as assume that S is ordinal scaled with equal distance to each other. This equal distance assumption if often not the case for grades or the RA assessment. We did not change this statement in the manuscript.

Point 16:

Line 64 does not make sense. Do you mean ‘RPNs with the same scores but derived differently?’

Answer: Thank you for that comment. We adjusted the suggested sentence in the manuscript, please consider line 64

Point 17:

Line 126/127 – Change to ‘We assume the process expert evaluates the change of each PP within the judged range.

Answer: Thank you for that comment. We adjusted the suggested sentence accordingly. Please consider line 128+129

Point 18:

Line 131 – reference missing

Answer: Thank you for that comment. We added the reference to the manuscript of Zahel et al. (2017) where the integrated process modeling approach was described for the first time.

Point 19:

Line 152 – reference missing

Answer: Thank you for that comment. We partly agree with that comment. Since the critical CQA is process related based on process knowledge or agency regulation, we assumed that this is not a case where a reference is needed since is general for each process. If we start to refer to a particular paper, we have to consider also multiple different publication, with different critically assessment approaches. Which is out of scope and we did not add a reference here.

Point20:

Line 182 – this should be figure 2 as figure 1 appears on line 90. Need to check text where you are referencing to figure 1 and figure 2

Answer: Thank you for that comment. During submission a problem with the figure numbering occurred. We fixed this in the entire manuscript.

Point 21:

Line 200 – reference missing

Answer: Thank you for that comment. We added that the required inputs are need based on the used python package.

Point 22:

Line 332 – change ‘reject’ to ‘rejected’

Answer: Thank you for that comment. We adjusted the suggested sentence like “we have to reject” instead of “rejected” as suggested. Please consider Line 334.

Point 23:

Line 384/385: We propose a new RA evaluation approach which avoids the limitations of using RPNs and relates the risk quantitatively to the probability of drug substance CQAs being out of specification

Answer: Thank you for that comment. We adjust the sentence according your suggestion. Please consider line 387 and 388

Point 24:

Line 434 – reference missing

Answer: Thank you for that comment we added references to that statement.

Point 25:

Line 441 and 442 – Scala? Do you mean Scale?

Answer: Thank you for that comment. This was a typo. We changed to Scale.